

# Long-term administration of vitamin B12 and adenosine triphosphate for idiopathic sudden sensorineural hearing loss: a retrospective study

Takaomi Kurioka[1], Hajime Sano[2], Shogo Furuki[1] and Taku Yamashita[1]

[1] Department of Otorhinolaryngology, Head and Neck Surgery, Kitasato University, Kanagawa, Japan
[2] School of Allied Health Sciences, Kitasato University, Kanagawa, Japan

## ABSTRACT

**Background.** In idiopathic sudden sensorineural hearing loss (ISSNHL), the relationship between the administration duration of vitamin B12 (vit B12) with adenosine triphosphate (ATP) and their therapeutic effect is not fully understood.

**Objective.** To investigate the therapeutic effect of long-term 16 ($\geq$weeks) administration of vit B12 with ATP on the prognosis of ISSNHL patients and compare it with those of short-term ($<8$ weeks) and middle-term ($\geq8$ weeks, $<16$ weeks) administration.

**Methods.** We retrospectively reviewed the medical records of 117 patients with ISSNHL treated between 2015 and 2018.

**Results.** The overall recovery rate was 32.5%. Initial higher hearing threshold and initial higher grade of hearing loss (HL) were associated with a poor prognosis. However, the administration duration of vit B12 and ATP did not influence the overall hearing improvement. With regard to the time course of hearing recovery, there was no significant difference in hearing recovery among the long-, middle-, and short-term administration groups until 16 weeks after treatment. However, at 16–24 weeks after initial treatment, the short-term administration group exhibited significantly lower hearing recovery than did the long-term administration groups.

**Conclusions.** The administration duration of vit B12 and ATP did not influence the overall hearing prognosis in ISSNHL, but long-term administration of vit B12 and ATP helped prevent the progression of HL after ISSNHL. Our results suggest that long-term administration of vit B12 and ATP is not necessarily required to treat ISSNHL patients, except for slowly progressing HL in the affected ears.

## INTRODUCTION

Idiopathic sudden sensorineural hearing loss (ISSNHL) usually occurs as an acute unilateral hearing loss (HL) (*Staecker et al., 2019*). The exact cause of ISSNHL has not been identified, but several possible etiologies have been proposed, such as vascular dysfunction, neurological disorder, autoimmune disease, or viral infection (*Watanabe et al., 2019*). Accordingly, the optimal treatment modality for ISSNHL remains controversial, but

Corresponding author
Takaomi Kurioka,
takaomi@xj9.so-net.ne.jp

various comprehensive approaches have been widely adopted, including topical application of steroids, vasodilator, and vitamin supplementation, with systemic steroid therapy being the mainstream treatment (*Ahmadzai et al., 2019*). However, despite systemic treatment, the spontaneous recovery rate only ranges from 30% to 60% (*Qiang et al., 2017*). Further, although steroid treatment has profound therapeutic benefits, it does not confer therapeutic benefit if the treatment is started more than 2 weeks after onset (*Amarillo et al., 2019*). After completion of the initial steroid therapy, patients are prescribed various medications such as vitamin supplementation and vasodilator drugs continuously. Particularly, a combination of vitamin B12 (vit B12) and adenosine triphosphate (ATP) is prescribed for ISSNHL patients to improve the auditory neural function and cochlear blood circulation (*Ahmadzai et al., 2019*).

The duration of vit B12 and ATP treatment depends on the period during which the treatment effect of these drugs is sustained. A patient would be willing to take vit B12 and ATP orally for the long term if these medications are likely to provide benefits for a longer period of time. However, the influence of the duration of vit B12 treatment with ATP on hearing prognosis is unclear. Thus, the present study aimed to investigate the influence of treatment duration of vit B12 with ATP on the hearing prognosis of ISSNHL. Towards this goal, we evaluated the therapeutic effect of long-term ($\geq$16 weeks) administration of vit B12 with ATP in comparison to that of short-term (<8 weeks) and middle-term ($\geq$8 weeks, <16 weeks) administration.

## MATERIALS AND METHODS

### Study design and ethical considerations

This retrospective study (B20-056) was approved by the Institutional Review Board of Kitasato University Hospital. The need for informed consent was waived owing to the retrospective nature of the study.

### Patients

The subjects were 117 patients with ISSNHL who underwent pure-tone audiometry at Kitasato University Hospital between 2015 and 2018. The inclusion criteria were as follows: (1) sudden sensorineural HL of 30 dB or greater in at least three consecutive frequencies; (2) underwent early therapeutic management, that is, treatment was started within 2 weeks after onset; (3) age over 18 years; and (4) follow-up duration longer than 6 months. The exclusion criteria were as follows: (1) age younger than 17 years; (2) middle ear or retro-cochlear pathology; (3) history of Meniere's disease or autoimmune HL; (4) history of genetic or fluctuating HL; (5) history of hearing aid wearing or otologic surgery; and (6) history of intratympanic steroid injection for salvage treatment after systemic steroid administration.

### Assessment of hearing function

Pure-tone audiometry was performed using a conventional device (AA-78; Rion, Tokyo, Japan) in a soundproof room. First, the hearing thresholds were obtained through air conduction (AC) and bone conduction (BC) in frequencies of 0.25, 0.5, 1, 2, and 4 kHz

**Table 1  Final outcomes according to Siegel's criteria.**

| Description | $n$ (%) |
|---|---|
| Complete recovery: final hearing level <25 dB | 23 (19.7) |
| Partial recovery: final hearing level 25–45 dB with hearing gain ≥ 15 dB | 15 (12.8) |
| Slight recovery: final hearing >45 dB with hearing gain ≥ 15 dB | 46 (39.3) |
| No recovery: hearing gain <15 dB | 33 (28.2) |

for both ears. To prevent cross-hearing phenomenon causing an erroneous measurement, masking noise was used to occupy the non-test ear while the other ear was tested, as necessary. Briefly, the necessity of masking during the measurement of the AC thresholds was identified based on the minimum interaural attenuation level of 40 dB for retesting through AC. Meanwhile, in the measurement of BC thresholds, a masking process was applied using ABC methods (*Kurioka et al., 2020b*). Thresholds were obtained across all the frequency octaves from 0.25 kHz to 4 kHz, and the arithmetic average AC and BC thresholds were calculated from the thresholds at 0.25, 0.5, 1, 2, and 4 kHz. The grade of HL, defined according to the Japanese Ministry of Health and Welfare guidelines (*Nakashima et al., 2014*), was then determined using the initial audiogram data. Hearing recovery was calculated as the difference between average hearing thresholds at different time points, including initial day of treatment and 2, 4, and 6 months after initial treatment. Siegel's criteria were employed to assess treatment results 6 months after treatment (Table 1), and the patients were accordingly classified into two groups as the recovery group (i.e., complete and partial recovery) and the no recovery group (i.e., slight recovery and no improvement). These two groups were treated with same drug regimen, except for the duration of vit B12 and ATP administration.

## Treatment

All patients were treated with a 10-day course of systemic corticosteroids (betamethasone 8 mg via intramuscular injection for the first day followed by betamethasone 4 mg via oral administration for the first 3 days, tapered to 2 mg for the second 3 days, and finally to 1 mg for the last 3 days). Oral administration of vit B12 (1.5 mg daily) and ATP (300 mg daily) was started from initial day of treatment, concurrent with the corticosteroids. To investigate the therapeutic effects of long-term administration of vit B12 and ATP on hearing outcomes in ISSNHL, the patients were divided into three groups according to the administration duration as the short-term (<8 weeks), middle-term (≥8 weeks, <16 weeks) and long-term (≥16 weeks) administration groups.

## Statistical analyses

The Chi-squared test was used to evaluate clinical characteristics and possible prognostic factors. Student's t-tests and non-parametric Mann–Whitney U tests were used to investigate continuous prognostic factors. For comparisons between more than two groups, a one-way analysis of variance was used followed by Dunn's multiple comparisons for the post hoc test. The parameters that were statistically significant in the univariate
**Table 2  Clinicodemographic patient characteristics.**

| Variables | Value |
|---|---|
| Number of patients | 117 |
| Age (years) | 58.2 ± 16.6 |
| Sex (male/female) | 66/51 |
| Initiation of treatment (days) | 4.2 ± 3.8 |
| Vertigo (±) | 56/61 |
| Diabetes (±) | 32/85 |
| Initial hearing level (dB) | 80.7 ± 24.0 |
| Initial grade of HL (1/2/3/4) | 8/18/45/46 |
| Duration of vit B12 and ATP administration (months) | 4.2 ± 2.4 |

analysis were entered into a binary logistic regression model for multivariate analysis. All statistical analyses were performed using GraphPad Prism 8 (GraphPad Software Inc., La Jolla, CA) or JMP 14.2 (SAS Institute Japan Inc., Tokyo, Japan). A $p$ value of <0.05 was considered statistically significant.

## RESULTS

### Patient characteristics and treatment outcome

The mean age was 58.2 ± 16.6 years, and there were 66 and 51 male and female patients, respectively. The patients' clinicodemographic characteristics are presented in Table 2. The mean interval between symptom onset and initial treatment was 4.2 ± 3.8 days. As accompanying symptoms and complications, 56 patients (47.9%) had vertigo and 32 patients (27.4%) had diabetes. The mean hearing threshold at the initial examination was 80.7 ± 24.0 dB. With respect to initial HL grade, 8 patients had grade 1; 18 patients, grade 2; 45 patients, grade 3; and 46 patients, grade 4. The average duration of vit B12 and ATP administration was 4.2 ± 2.4 months. With regard to final recovery according to Siegel's criteria, the overall recovery rate (complete + partial recovery) was 32.5%. Specifically, 19.7%, 12.8%, 39.3%, and 28.2% of the patients achieved complete recovery, partial recovery, slight recovery, and no improvement, respectively (Table 1).

### Prognostic factors

In univariate analysis, the initial hearing threshold and initial grade of HL was significantly lower in the recovery group than that in the no recovery group ($p < 0.0001$). With regard to the associated symptoms, there was a significantly higher incidence of vertigo in the no recovery group than in the recovery group ($p = 0.04$). The other variables, including age, sex, days to initiation of treatment, and diabetes were not significantly different between the two groups. The duration of vit B12 and ATP administration was slightly longer in the no recovery group than that in the recovery group, but the difference was not significant. The significant variables in the univariate analysis were included in the multivariate analysis. The results showed that higher initial hearing threshold and higher initial grade of HL were associated with a poor prognosis in ISSNHL patients (Table 3).

**Table 3  Prognostic factors of ISSNHL.**

| | Recovery group ($n = 38$) | No recovery group ($n = 79$) | Univariate | Multivariate |
|---|---|---|---|---|
| | | | $p$ | $p$ |
| Age (years) | 55.6 ± 15.5 | 59.5 ± 17.0 | 0.23 | |
| Sex (male/female) | 17/21 | 49/30 | 0.08 | |
| Time to initiation of treatment (days) | 4.3 ± 3.0 | 4.2 ± 4.2 | 0.90 | |
| Vertigo (±) | 13/25 | 43/36 | 0.04* | 0.27 |
| Diabetes (±) | 12/26 | 22/57 | 0.68 | |
| Initial hearing level (dB) | 65.9 ± 17.7 | 87.9 ± 23.4 | <0.0001**** | 0.001** |
| Initial grade of HL (1/2/3/4) | 3/11/22/2 | 5/7/23/44 | <0.0001**** | 0.03* |
| Duration of vit B12 and ATP administration (months) | 3.7 ± 2.6 | 4.4 ± 2.2 | 0.11 | |

## Administration duration of vit B12 and ATP

The short-term, middle-term, and long-term administration groups involved 15, 41, and 38 patients, respectively. The patients with complete recovery within 24 weeks after onset ($n = 23$) were excluded because the vit B12 and ATP administration was terminated following the confirmation of complete recovery. Specifically, the duration of drug administration was shortened due to complete recovery, but the shortened administration did not contribute to complete recovery. Therefore, it was not appropriate to include the patients with complete recovery within 24 weeks after onset for the analysis of the effect of drug administration duration. In this study, the duration of administration was determined by the patients after consultation with their physician, even though they did not achieve complete recovery. The clinical features of each group are presented in Table 4. The mean duration of drug administration in the short, middle, and long-term groups was 1.9, 3.3, and 6.3 months, respectively. In univariate analysis, the duration of drug administration was significantly different among the groups ($p < 0.0001$), but the clinical characteristics and hearing results were comparable ($p > 0.05$). This indicated that the duration of drug administration was not influenced by the patient's clinical characteristics and did not affect the overall hearing outcome. Ultimately, these results show that long-term administration of vit B12 and ATP had no significant impact on the hearing prognosis within our observation period.

We also investigated the time course of hearing recovery in each group. As shown in Fig. 1A, the thresholds (dB) of total hearing recovery within 24 months after symptom onset in the short-, middle-, and long-term administration groups were 18.9 ± 5.8, 22.4 ± 2.3, and 22.7 ± 3.0, respectively, with no significant differences ($p = 0.76$). The time course of hearing recovery in the overall population was 18.9 ± 1.7, 3.1 ± 0.7, and 0.04 ± 0.4 within 0–2, 2–4, and 4–6 months after symptom onset, respectively ($p < 0.0001$; Fig. 1B). Hearing recovery within the initial 2 months after symptom onset was significantly higher than that within 2–4 months ($p < 0.0001$) and 4–6 months ($p < 0.0001$). Moreover, hearing recovery within 2–4 months after symptom onset was significantly higher than that

**Table 4** Clinicodemographic patient characteristics according to the duration of vit B12 and ATP administration.

| | Short (n = 15) | Medium (n = 41) | Long (n = 38) | p |
|---|---|---|---|---|
| Duration of vit B12 and ATP administration (months) | 1.9 ± 0.4 | 3.3 ± 0.9 | 6.3 ± 1.8 | <0.0001**** |
| Age (years) | 65.7 ± 16.6 | 60.2 ± 14.4 | 56.4 ± 17.7 | 0.17 |
| Sex (male/female) | 12/3 | 20/21 | 24/14 | 0.09 |
| Time to initiation of treatment (days) | 4.7 ± 6.1 | 3.9 ± 3.1 | 4.4 ± 4.3 | 0.78 |
| Vertigo (±) | 7/8 | 19/22 | 21/17 | 0.70 |
| Diabetes (±) | 3/12 | 14/27 | 11/27 | 0.59 |
| Initial hearing level (dB) | 87.7 ± 23.8 | 80.1 ± 22.7 | 89.2 ± 22.9 | 0.19 |
| Initial grade of HL (1/2/3/4) | 1/1/4/9 | 2/7/17/15 | 2/4/11/21 | 0.62 |
| Final hearing level (dB) | 68.8 ± 29.8 | 57.8 ± 21.3 | 66.5 ± 20.7 | 0.14 |
| Hearing improvement (Type 2/3/4) | 3/5/7 | 10/19/12 | 2/22/14 | 0.13 |

within 4–6 months ($p = 0.04$). This result indicated that hearing recovery in the initial 2 months after onset is higher and that the probability of recovery decreases as the duration after symptom onset increases. The time course of hearing recovery of both the affected and contralateral ears in the three groups are shown in Fig. 2. There was no significant difference in hearing recovery among the three groups until 16 weeks after onset. However, hearing recovery of the affected ears was significantly lower in the short-term administration group than that in the long-term administration group at 16–24 weeks after symptom onset ($p = 0.03$). Meanwhile, there was no significant difference in hearing recovery of the contralateral ears among the three groups throughout the study period. This indicated a slow progression of HL in the affected ear and that long-term administration of vit B12 and ATP might prevent progressive HL in the affected ear in ISSNHL. We performed an additional analysis of hearing recovery at each frequency. As shown in Fig. 3, hearing recovery appeared to be greater at 0.5 and 1.0 kHz than that at 2.0 and 4.0 kHz until 8 weeks after onset. In addition, significant differences between long-term and short-term drug administration were observed at only 2.0 kHz during the 4–6 months after the onset of ISSNHL.

## DISCUSSION

The influence of the duration of vit B12 and ATP treatment on hearing prognosis has been unclear. In our study, the administration periods of vit B12 and ATP were not prognostic factors. However, this result does not necessarily indicate that long-term administration of vit B12 and ATP has no therapeutic effect on the overall prognosis of ISSNHL as hearing recovery was significantly lower in the short-term administration group than that in the middle and long-term administration groups within 4–6 months after onset. This indicates that vit B12 and ATP treatment might have gradual therapeutic effects on preventing the deterioration in hearing rather than promoting the hearing recovery. Hearing recovery within the initial 2 months was also significantly higher than that in the later phase,

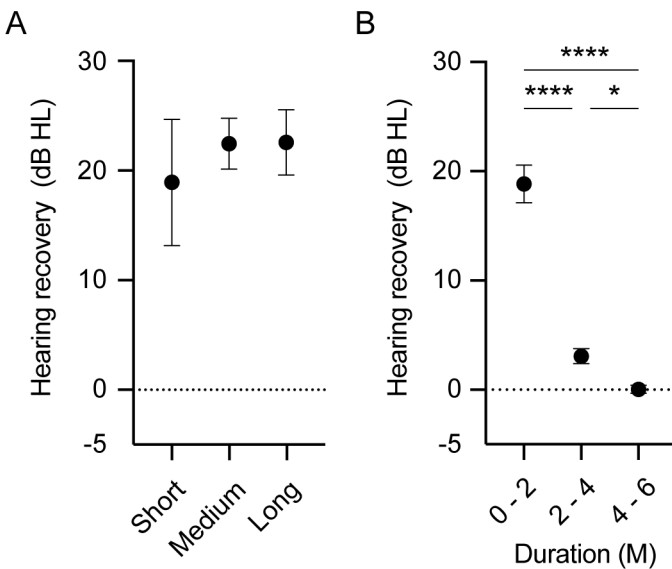

**Figure 1** **Overall hearing recovery according to the duration of vit B12 and ATP administration (A) and in the overall population (B).** (A) The total hearing threshold recovery from symptom onset to 6 months was comparable among the three groups. (B) Hearing recovery in the initial 2 months was significantly higher than that in the later periods. Hearing recovery within 2–4 months after symptom onset was also significantly higher than that at 4–6 months after onset. $p$ values are indicated as $**p < 0.01$, $****p < 0.0001$.

indicating that early initial treatment within 2 months after symptom onset is extremely important.

The optimal treatment strategy for ISSNHL has not been established to date because the cause and etiology of ISSNHL remain unclear. However, circulation and neurological disorders are thought to be one of the major causes for ISSNHL (*Hsu et al., 2016*). Because the cochlea is supplied by the labyrinthine artery, which has no collateral circulation, the obstruction of blood supply by thrombosis or hemorrhage may cause cochlear damages that results in ISSNHL. Diabetes, hyperlipidemia, and aging are also well-known factors of microvascular disease that results in blood circulation disorders in the inner ear (*Akinpelu et al., 2014*; *Orita et al., 2007*). Various empirical treatments have been applied to improve blood circulation and to increase oxygen supply to the inner ear, including vasodilators, steroids, plasma expanders, and anticoagulant agents (*Xie et al., 2018*). Particularly, systemic steroids have become the most commonly used treatment for ISSNHL patients, with the dose tapered within 2 weeks (*Slattery et al., 2005*). The use of intra-tympanic steroid injection for the treatment of ISSNHL has also recently increased owing to its comparable effects to systemic steroids and the additional benefit of salvage treatment (*Kordis & Battelino, 2017*). After the initial steroid treatment, which usually lasts for 1–2 weeks, patients are generally prescribed vit B12 and ATP supplementation for further management.

Vitamin B12 plays an important role in the maintenance of normal neural function (*Altun & Kurutas, 2016*). Accordingly, vit B12 deficiency leads to anemia, demyelination,

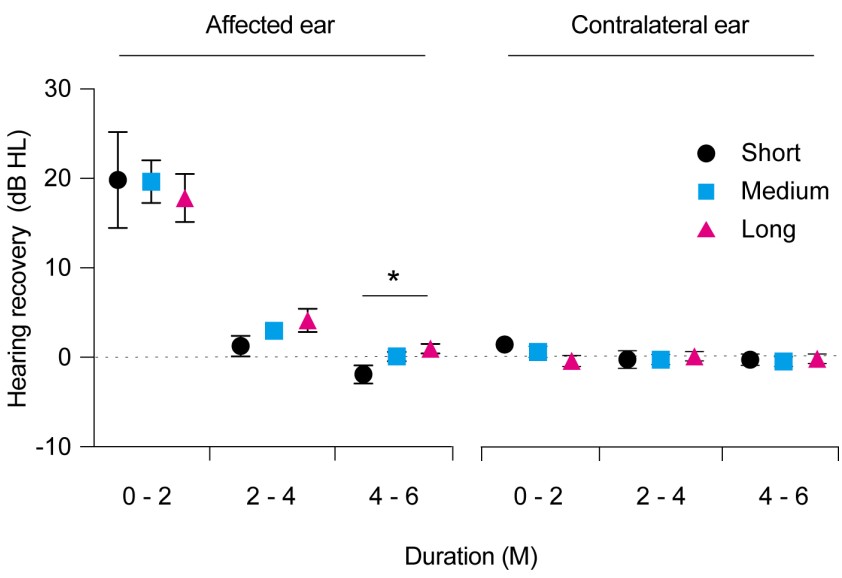

**Figure 2** **Time course of hearing recovery in the affected ears and contralateral ears.** Hearing recovery in the affected ears was comparable among the three groups within 0–2 and 2–4 months after symptom onset. Hearing recovery in the short-term administration group (black) was significantly lower than that in long-term administration groups (red) at 4–6 months after symptom onset. In the contralateral ears, there were no significant differences in hearing recovery among the three groups across all time points after symptom onset. $p$ values are indicated as *$p < 0.05$.

axonal degeneration, and, ultimately, neuronal loss. In the auditory system, vit B12 deficiency is known to have negative effects on hearing by affecting myelinization of the auditory neurons at the retro-cochlear region. Additionally, auditory neural degenerations, such as disruption of cochlear synapses, demyelination, and shrinkage of auditory nerves, gradually progress after decreased auditory inputs (*Kurioka et al., 2020a*; *Kurioka, Mogi & Yamashita, 2020*). Vitamin B12 supplementation has been shown to increase the number of Schwann cells and myelinated nerve fibers and the diameter of the axons, thereby promoting the regeneration of myelinated nerve fibers and the proliferation of Schwann cells (*Lopatina et al., 2011*). Therefore, supplementary vit B12 treatment is considered in auditory diseases to prevent neural disorders and to enhance neurovascular endothelial function (*Singh et al., 2016*). Vitamin B12 supplementation also reduces homocysteine synthesis, which is a vascular and thrombotic risk factor and causes vascular injury by reducing the amount of nitric oxide (NO). The reduced homocysteine synthesis leads to vasodilatation as a result of an increase in the amount of NO (*Toda & Okamura, 2016*). Thus, vit B12 can cause an increase in vascular perfusion and cellular metabolism in the cochlea.

ATPs are often used as vasodilators to increase cochlear blood flow in auditory diseases. The vasodilator effect of ATP is mediated by the endothelium and follows the release of NO. A previous study reported that treatment with ATP resulted in a large increase in cochlear blood flow and a decrease in blood pressure owing to the vasodilator actions of ATP (*Munoz, McFie & Thorne, 1999*). Another study reported that a 300 mg dose of ATP

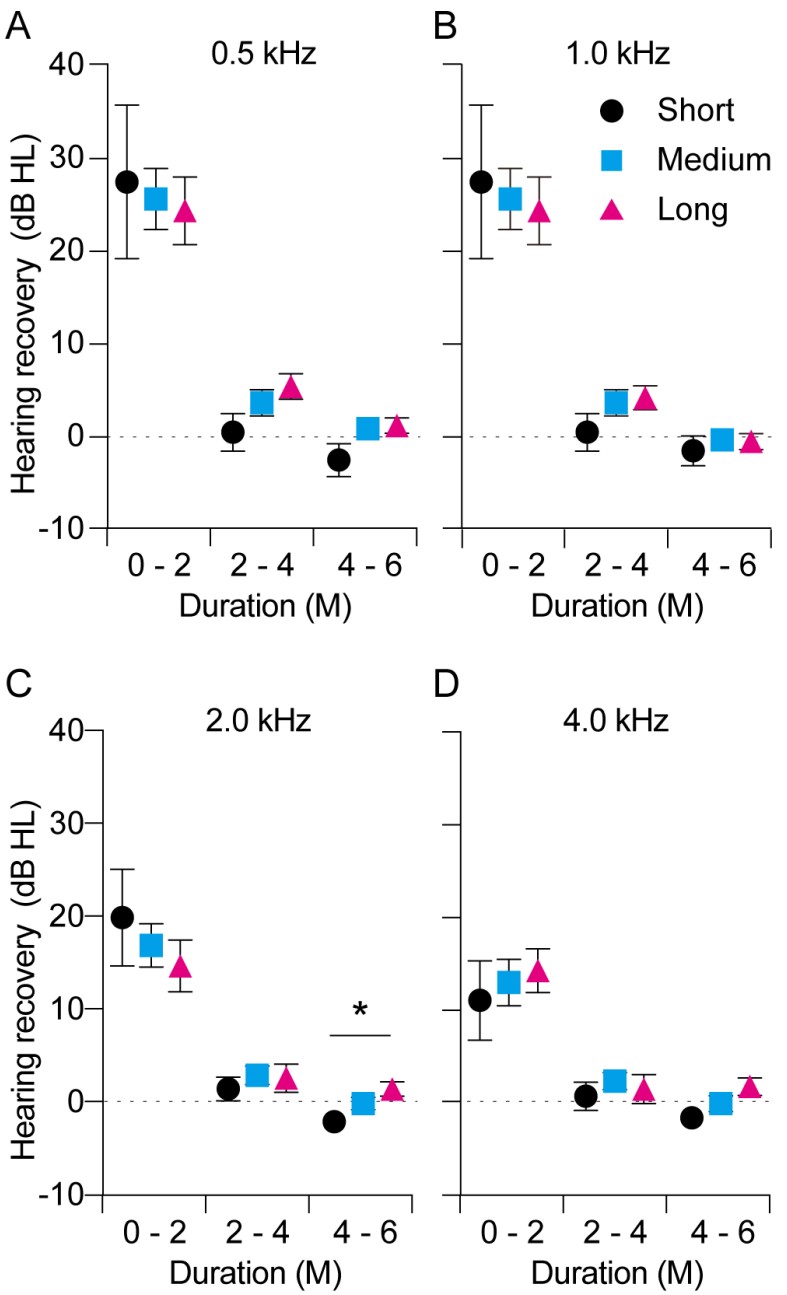

**Figure 3** **Time course of hearing recovery in the affected ears at each frequency over the 0.5–4.0 kHz range.** Hearing recovery in the affected ears was comparable among the three groups within 0–2 and 2–4 months after onset. Hearing recovery differed significantly between the short-term (black) and long-term administration groups (red) at 4–6 months after onset only at 2.0 kHz (C). *P* values are indicated as *$p <$ 0.05.

has therapeutic effects for inner ear pathologies, such as Meniere's disease (*Mizukoshi et al., 1983*). Thus, ATP and vit B12 are widely used for the treatment of ISSNHL owing to their various biological therapeutic effects.

The prognosis for recovery from ISSNHL can be influenced by various factors, including age, accompanying symptoms such as vertigo, degree of HL, audiometric configuration, and time to treatment initiation (*Kim et al., 2018*; *Kuhn et al., 2011*; *Lin et al., 2016*). The findings of the present study indicated that the initial hearing threshold and initial grade of HL were useful prognostic factors, consistent with the findings of a previous study (*Kuhn et al., 2011*). With respect to the treatment period after ISSNHL onset, it has been reported that hearing remained relatively stable after a period of 2–3 months from symptom onset (*Kallinen et al., 2001*; *Kanzaki, Taiji & Ogawa, 1988*; *Psifidis, Psillas & Daniilidis, 2006*). It is possible that the period of 2 months could be consistent with the natural history of the disease, regardless of which therapeutic strategy is applied. Thus, any additional treatment after 2 months should not affect the outcome of the hearing, consistent with our results. However, a long-term follow-up study reported that 9.9% patients showed an improvement over a period of 3 months after the onset of ISSNHL (*Yeo et al., 2007*). Nevertheless, one study found that 25% of patients had long-term hearing deterioration after the onset of ISSNHL (*Furuhashi et al., 2002*). This is consistent with our results that ISSNHL affected the involved ear more gradually than the contralateral ears. This hearing deterioration appeared to have no relationship with age-related HL because the deterioration was observed only in the affected ears in this study. Although the overall hearing recovery did not differ according to the duration of administration, we found some benefit for the prevention of hearing deterioration with long-term administration. However, these results generally do not indicate that long-term vit B12 and ATP should be adopted for ISSNHL patients. Furthermore, it is generally believed that hearing recovery would differ between acute low-tone sensorineural HL and ISSNHL. Therefore, the initial type of HL in ISSNHL might be associated with delayed changes in hearing after the onset. Additional studies with a longer observation period are needed to elucidate the accurate therapeutic effects of vit B12 and ATP in ISSNHL.

This study has some limitations. First, this was a retrospective study conducted in a single hospital, and the sample size was relatively small. Second, the duration of vit B12 and ATP treatment in this study was influenced by various factors, such as patient preferences and drug therapeutic effects. However, despite these limitations, we believe that our study is valuable because to the best of our knowledge, this is the first to focus on the relationship between the duration of vit B12 and ATP administration and prognosis in ISSNHL. Our findings have important clinical implications in the understanding and management of patients with ISSNHL. Although the underlying therapeutic mechanisms of vit B12 and ATP in ISSNHL are not well known, its effects need to be considered in the management of ISSNHL in the clinic. Further prospective studies with larger populations are needed to validate our findings.

## CONCLUSIONS

The administration duration of vit B12 and ATP did not influence the hearing prognosis in ISSNHL. Higher initial hearing threshold and higher initial grade of HL were associated with a poor prognosis. Compared to the contralateral ear, HL progressed gradually in the affected ear. Long-term administration of vit B12 and ATP might help prevent progressive HL in the affected ear.

### Funding

This work was supported by a GSK Japan Research Grant 2019, JSPS KAKENHI grant (Grant Number 19K24052) and a JSPS KAKENHI grant (Grant Number 20K18263) (all to Takaomi Kurioka). The funders had no role in study design, data collection and analysis, decision to publish, or preparation of the manuscript.

### Grant Disclosures

The following grant information was disclosed by the authors:
GSK Japan Research Grant 2019.
JSPS KAKENHI grant: 19K24052, 20K18263.

### Competing Interests

The authors declare there are no competing interests.

### Author Contributions

- Takaomi Kurioka conceived and designed the experiments, performed the experiments, analyzed the data, prepared figures and/or tables, authored or reviewed drafts of the paper, and approved the final draft.
- Hajime Sano, Shogo Furuki and Taku Yamashita performed the experiments, analyzed the data, prepared figures and/or tables, and approved the final draft.

### Ethics

The following information was supplied relating to ethical approvals (i.e., approving body and any reference numbers):

All protocols performed in studies were approved by the Institutional Review Board at Kitasato University Hospital (B20-056).

### Data Availability

The raw data is available in the Supplementary File.

### Supplemental Information

Supplemental information for this article can be found online at http://dx.doi.org/10.7717/peerj.10406#supplemental-information.

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
