# Peer review of "Long-term administration of vitamin B12 and adenosine triphosphate for idiopathic sudden sensorineural hearing loss: a retrospective study"

_PeerJ, doi:10.7717/peerj.10406_

## Round 0.1 · original submission · Minor Revisions

Clearly the reviewers were happy with the overall article, although it would be helpful if you could address the comments of Reviewer 1 around the details of the nature of the ISSNHL, if you have the information.

I would also like to see some more background on the use of ATP as a treatment ... my understanding is that ATP at this dose would not be significantly bio-available, and while the paper cited on guinea pig cochlear blood flow and ATP is certainly relevant preclinical evidence, is there any evidence in people that might speak to how ATP might affect recovery from/progression of hearing loss ?

Reviewer 1 ·

Basic reporting

no comments

Experimental design

no comments

Validity of the findings

no comments

Additional comments

This manuscript described that long-term administration of vit B12 & ATP helped prevent the progression of hearing loss after idiopathic sudden sensorineural hearing loss (ISSNHL).
Figure 2 revealed that hearing outcome was better during 4 – 6 months after the onset of ISSNHL in the group taking vit B12 & ATP than that not taking vit B12 & ATP. Although the difference was small, it was significant statistically.

Because many otologists consider that hearing level became stable after three months from the onset of ISSNHL, more detailed description is required.

#1
My understanding is that hearing level moves longer in low-tone frequencies that in high-tone frequencies after the onset of ISSNHL. What was the frequency area of the slowly progressing hearing loss during 4 – 6 months after the onset of ISSNHL?

#2
It is generally considered that hearing recovery can be expected longer in acute low-tone SNHL than in ordinary ISSNHL. I expect some comments about the initial hearing type of ISSNHL in association with the delayed hearing change after the onset of ISSNHL.

#3
Furuhashi’s paper (Clinical Otolaryngol, 2002) described recurrence of ISSNHL for a very long period like ten years.
I would like to recommend a paper investigating for more than three months after the onset of ISSNHL for one of the references.
Yeo, SW, Lee DH et al.: Hearing outcome of sudden sensorineural hearing loss: Long-term follow-up. Otolaryngol Head Neck Surg 136;221-224. 2007.

·

Basic reporting

no comment

Experimental design

no comment

Validity of the findings

no comment

Additional comments

Vitamin (vit) B12 and adenosine triphosphate (ATP) are usually used for the treatment of idiopathic sudden sensorineural hearing loss. However, we don’t know how long these medicines should be used. This paper clearly answers this question. In this paper, authors clarified that vit B12 and ATP did not influence the overall prognosis of idiopathic sudden sensorineural hearing loss; however, the long-term administration can prevent the progression of hearing loss after idiopathic sudden sensorineural hearing loss. I think that this must provide clinically useful information to the readers of PeerJ. Actually, the number of enrolled subjects is not many (n=117), but this issue is mentioned as a limitation of this study. Therefore, I would like to recommend this paper for the publication.

---

## Round 0.2 · accepted · Accept

Thank you for addressing the comments of reviewers, and including data that further elaborates on your findings.